# MULTIGRAPH TOPOLOGY DESIGN FOR CROSS-SILO FEDERATED LEARNING

## ABSTRACT

Cross-silo federated learning utilizes a few hundred reliable data silos with high-speed access links to jointly train a model. While this approach becomes a popular setting in federated learning, designing a robust topology to reduce the training time is still an open problem. In this paper, we present a new multigraph topology for cross-silo federated learning. We first construct the multigraph using the overlay graph. We then parse this multigraph into different simple graphs with isolated nodes. The existence of isolated nodes allows us to perform model aggregation without waiting for other nodes, hence reducing the training time. We further propose a new distributed learning algorithm to use with our multigraph topology. The intensive experiments on public datasets show that our proposed method significantly reduces the training time compared with recent state-of-the-art topologies while ensuring convergence and maintaining the accuracy.

## 1 INTRODUCTION

Federated learning entails training models via remote devices or siloed data centers while keeping data locally to respect the user's privacy policy (Li et al., 2020a). According to Kairouz et al. (2019), there are two popular training scenarios: the *cross-device* scenario, which encompasses a variety (millions or even billions) of unreliable edge devices with limited computational capacity and slow connection speeds; and the *cross-silo* scenario, which involves only a few hundred reliable data silos with powerful computing resources and high-speed access links. Recently, cross-silo scenario becomes popular in different federated learning applications such as healthcare (Xu et al., 2021), robotics (Nguyen et al., 2021; Zhang et al., 2021c), medical imaging (Courtiol et al., 2019; Liu et al., 2021), and finance (Shingi, 2020).

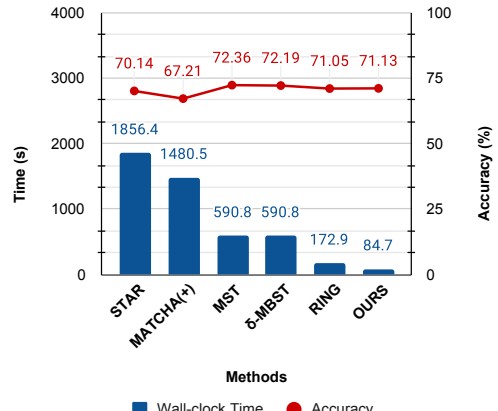

Figure 1: Comparison between different topologies on FEMNIST dataset and Exodus network (Miller et al., 2010). The accuracy and total wall-clock training time (or overhead time) are reported after $6,400$ communication rounds.

In practice, federated learning is a promising research direction where we can utilize the effectiveness of machine learning methods while respecting the user's privacy. Key challenges in federated learning include model convergence, communication congestion, and imbalance of data distributions in different silos (Kairouz et al., 2019). A popular federated training method is to set a central node that orchestrates the training process and aggregates contributions of all clients. Main limitation of this client-server approach is that the server node potentially represents a communication congestion point in the system, especially when the number of clients is large. To overcome this limitation, recent research has investigated the decentralized (or peer-to-peer) federated learning approach. In the aforementioned approach, the communication is done via peer-to-peer topology without the need for a central node. However, main challenge of decentralized federated learning is to achieve fast training time, while assuring model convergence and maintaining the model accuracy.

In federated learning, the communication topology plays an important role. A more efficient topology leads to quicker convergence and reduces the training time, quantifying by the worst-case convergence bounds in the topology design (Jiang et al., 2017; Nedić et al., 2018; Wang & Joshi, 2018). Furthermore, topology design is directly related to other problems during the training process such as network congestion, the overall accuracy of the trained model, or energy usage (Yang et al., 2021; Nguyen et al., 2021; Kang et al., 2019). Designing a robust topology that can reduce the training time while maintaining the model accuracy is still an open problem in federated learning (Kairouz et al., 2019). This paper aims to design a new topology for cross-silo federated learning, which is one of the most common training scenarios in practice.

Recently, different topologies have been proposed for cross-silo federated learning. In (Brandes, 2008), the STAR topology is designed where the orchestrator averages all models throughout each communication round. Wang et al. (2019) propose MATCHA to decompose the set of possible communications into pairs of clients. At each communication round, they randomly select some pairs and allow them to transmit models. Marfoq et al. (2020) introduce the RING topology with the largest throughput using max-plus linear systems. While some progress has been made in the field, there are challenging problems that need to be addressed such as congestion at access links (Wang et al., 2019; Yang et al., 2021), straggler effect (Neglia et al., 2019; Park et al., 2021), or identical topology in all communication rounds (Jiang et al., 2017; Marfoq et al., 2020).

In this paper, we propose a new multigraph topology based on the recent RING topology (Marfoq et al., 2020) to reduce the training time for cross-silo federated learning. Our method first constructs the multigraph based on the overlay of RING topology. Then we parse this multigraph into simple graphs (i.e., graphs with only one edge between two nodes). We call each simple graph is a *state* of the multigraph. Each state of the multigraph may have isolated nodes, and these nodes can do model aggregation without waiting for other nodes. This strategy significantly reduces the cycle time in each communication round. To ensure model convergence, we also adapt and propose a new distributed learning algorithm. The intensive experiments show that our proposed topology significantly reduces the training time in cross-silo federated learning (See Figure 1).

## 2 LITERATURE REVIEW

**Federated Learning**. Federated learning has been regarded as a system capable of safeguarding data privacy (Konečnỳ et al., 2016; Gong et al., 2021; Zhang et al., 2021b; Li et al., 2021b). Contemporary federated learning has a centralized network design in which a central node receives gradients from the client nodes to update a global model. Early findings of federated learning research include the work of Konečnỳ et al. (2015), as well as a widely circulated article from McMahan & Ramage (2017). Then Yang et al. (2013); Shalev-Shwartz & Zhang (2013); Ma et al. (2015); Jaggi et al. (2014), and Smith et al. (2018) extend the concept of federated learning and its related distributed optimization algorithms. Federated Averaging (FedAvg) was proposed by McMahan et al. (2017), its variations such as FedSage (Zhang et al., 2021a) and DGA (Zhu et al., 2021b), or other recent state-of-the-art model aggregation methods (Hong et al., 2021; Ma et al., 2022; Zhang et al., 2022; Liu et al., 2022; Elgabli et al., 2022) are introduced to address the convergence and non-IID (non-identically and independently distributed) problem. Despite its simplicity, the client-server approach suffers from the communication and computational bottlenecks in the central node, especially when the number of clients is large (He et al., 2019; Qu et al., 2022).

**Decentralized Federated Learning**. Decentralized (or peer-to-peer) federated learning allows each silo data to interact with its neighbors directly without a central node (He et al., 2019). Due to its nature, decentralized federated learning does not have the communication congestion at the central node, however, optimizing a fully peer-to-peer network is a challenging task (Nedić & Olshevsky, 2014; Lian et al., 2017; He et al., 2018; Lian et al., 2018; Wang et al., 2019; Marfoq et al., 2020; 2021; Li et al., 2021a). Noticeably, the decentralized periodic averaging stochastic gradient descent (Wang & Joshi, 2018) is proved to converge at a comparable rate to the centralized algorithm while allowing large-scale model training (Wu et al., 2017; Shen et al., 2018; Odeyomi & Zaruba, 2021). Recently, systematic analysis of the decentralized federated learning has been explored by (Li et al., 2018b; Ghosh et al., 2020; Koloskova et al., 2020).

**Communication Topology**. The topology has a direct impact on the complexity and convergence of federated learning (Chen et al., 2020). Many works have been introduced to improve the effective-

ness of topology, including star-shaped topology (Brandes, 2008; Konečný et al., 2016; McMahan et al., 2016; 2017; Kairouz et al., 2019) and optimized-shaped topology (Neglia et al., 2019; Wang et al., 2019; Marfoq et al., 2020; Bellet et al., 2021; Vogels et al., 2021; Huang et al., 2022). Particularly, a spanning tree topology based on Prim (1957) algorithm was introduced by Marfoq et al. (2020) to reduce the training time. As mentioned by Brandes (2008), STAR topology is designed where an orchestrator averages model updates in each communication round. Wang et al. (2019) introduce MATCHA to speed up the training process through decomposition sampling. Since the duration of a communication round is dictated by stragglers effect (Karakus et al., 2017; Li et al., 2018a), Neglia et al. (2019) explore how to choose the degree of a regular topology. Marfoq et al. (2020) propose RING topology for cross-silo federated learning using the theory of max-plus linear systems. Recently, Huang et al. (2022) introduce Sample-induced Topology which is able to recover effectiveness of existing SGD-based algorithms along with their corresponding rates.

**Multigraph.** The definition of multigraph has been introduced as a traditional paradigm (Gibbons, 1985; Walker, 1992). A typical "graph" usually refers to a simple graph with no loops or multiple edges between two nodes. Different from a simple graph, multigraph allows multiple edges between two nodes. In deep learning, multigraph has been applied in different domains, including clustering (Martschat, 2013; Luo et al., 2020; Kang et al., 2020), medical image processing (Liu et al., 2018; Zhao et al., 2021; Bessadok et al., 2021), traffic flow prediction (Lv et al., 2020; Zhu et al., 2021a), activity recognition (Stikic et al., 2009), recommendation system (Tang et al., 2021), and cross-domain adaptation (Ouyang et al., 2019). In this paper, we construct the multigraph to enable isolated nodes and reduce the training time in cross-silo federated learning.

## 3 PRELIMINARIES

### 3.1 FEDERATED LEARNING

In federated learning, silos do not share their local data, but still periodically transmit model updates between them. Given $N$ siloed data centers, the objective function for federated learning is:

$$\min_{\mathbf{w} \in \mathbb{R}^d} \sum_{i=1}^{N} p_i \mathbb{E}_{\xi_i} \left[ L_i \left( \mathbf{w}, \xi_i \right) \right], \tag{1}$$

where $L_i(\mathbf{w}, \xi_i)$ is the loss of model $\mathbf{w} \in \mathbb{R}^d$. $\xi_i$ is an input sample drawn from data at silo $i$. The coefficient $p_i > 0$ specifies the relative importance of each silo. Recently, different distributed algorithms have been proposed to optimize Eq. 1 (Konecný et al., 2016; McMahan et al., 2017; Li et al., 2020b; Wang et al., 2019; Li et al., 2019; Wang & Joshi, 2018; Karimireddy et al., 2020). In this work, we use DPASGD (Wang & Joshi, 2018) algorithm to update the weight of silo $i$ in each training round as follows:

$$\mathbf{w}_i \left( k+1 \right) = \begin{cases} \sum_{j \in \mathcal{N}_i^+ \cup \{i\}} \mathbf{A}_{i,j} \mathbf{w}_j \left( k \right), & \text{if k} \equiv 0 \left( \bmod u+1 \right), \\ \mathbf{w}_i \left( k \right) - \alpha_k \frac{1}{b} \sum_{h=1}^{b} \nabla L_i \left( \mathbf{w}_i \left( k \right), \xi_i^{(h)} \left( k \right) \right), & \text{otherwise.} \end{cases} \tag{2}$$

where $b$ is the batch size, $i, j$ denote the silo, $u$ is the number of local updates, $\alpha_k > 0$ is a potentially varying learning rate at $k$-th round, $\mathbf{A} \in \mathbb{R}^{N \times N}$ is a consensus matrix with non-negative weights, and $\mathcal{N}_i^+$ is the in-neighbors set that silo $i$ has the connection to.

### 3.2 MULTIGRAPH FOR FEDERATED LEARNING

**Connectivity and Overlay**. Following Marfoq et al. (2020), we consider the *connectivity* $\mathcal{G}_c = (\mathcal{V}, \mathcal{E}_c)$ as a graph that captures possible direct communications among silos. Based on its definition, the connectivity is often a fully connected graph and is also a directed graph. The *overlay* $\mathcal{G}_o$ is a connected subgraph of the connectivity graph, i.e., $\mathcal{G}_o = (\mathcal{V}, \mathcal{E}_o)$, where $\mathcal{E}_o \subset \mathcal{E}_c$. Only nodes directly connected in the overlay graph $\mathcal{G}_o$ will exchange the messages during training. We refer the readers to Marfoq et al. (2020) for more in-deep discussions.

**Multigraph**. While the connectivity and overlay graph can represent different topologies for federated learning, one of their drawbacks is that there is only one connection between two nodes. In our

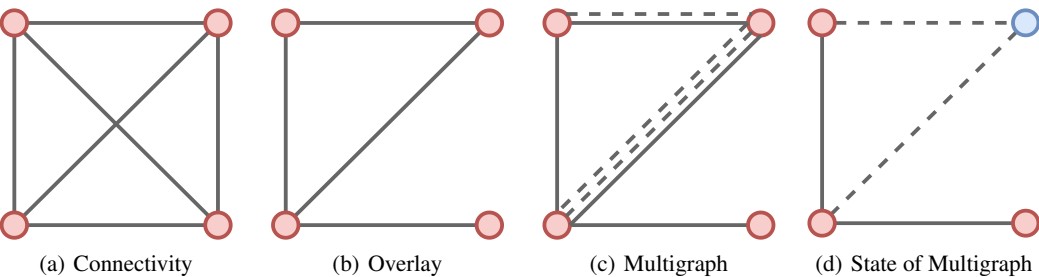

Figure 2: Example of connectivity, overlay, multigraph, and a state of our multigraph. Blue node is an isolated node. Dotted line denotes a weakly-connected edge.

work, we construct a *multigraph* $\mathcal{G}_m = (\mathcal{V}, \mathcal{E}_m)$ from the overlay $\mathcal{G}_o$. The multigraph can contain multiple edges between two nodes (Chartrand & Zhang, 2013). In practice, we parse this multigraph to different graph states, each state is a simple graph with only one edge between two nodes.

In the multigraph $\mathcal{G}_m$, the connection edge between two nodes has two types: strongly-connected edge and weakly-connected edge (Ke-xing et al., 2016). Under both strongly and weakly connections, the participated nodes can transmit their trained models to their out-neighbours $\mathcal{N}_i^-$ or download models from their in-neighbours $\mathcal{N}_i^+$. However, in a strongly-connected edge, two nodes in the graph must wait until all upload and download processes between them are finished to do model aggregation. On the other hand, in a weakly-connected edge, the model aggregation process in each node can be established whenever the previous training process is finished by leveraging up-to-dated models which have not been used before from the in-neighbours of that node.

**State of Multigraph**. Given a multigraph $\mathcal{G}_m$, we can parse this multigraph into different simple graphs with only one connection between two nodes (either strongly-connected or weakly-connected). We call each simple graph as a state $\mathcal{G}_m^s$ of the multigraph.

**Isolated Node**. A node is called isolated when all of its connections to other nodes are weakly-connected edges. The graph concepts and isolated nodes are shown in Figure 2.

### 3.3 DELAY AND CYCLE TIME IN MULTIGRAPH

**Delay**. Following Marfoq et al. (2020), a delay to an edge $e(i,j)$ is the time interval when node $j$ receives the weight sending by node $i$, which can be defined by:

$$d(i,j) = u \times T_c(i) + l(i,j) + \frac{M}{O(i,j)} \tag{3}$$

where $T_c(i)$ denotes the time to compute one local update of the model; $u$ is the number of local updates; $l(i,j)$ is the link latency; $M$ is the model size; $O(i,j)$ is the total network traffic capacity.

However, unlike other communication infrastructures, the multigraph only contains connections between silos without other nodes such as routers or amplifiers. Thus, the total network traffic capacity $O(i,j) = \min\left(\frac{C_{\text{UP}}(i)}{|\mathcal{N}_i^-|}, \frac{C_{\text{DN}}(j)}{|\mathcal{N}_i^+|}\right)$ where $C_{\text{UP}}$ and $C_{\text{DN}}$ denote the upload and download link capacity. Note that the upload and download can happen in parallel.

Since multigraph can contain multiple edges between two nodes, we extend the definition of the delay in Eq. 3 to $d_k(i,j)$, with $k$ is the $k$-th communication round during the training process, as:

$$d_k(i,j) = \begin{cases} d_k(i,j), & \text{if } (e_k(i,j) = \mathbb{1} \text{ and } e_{k-1}(i,j) = \mathbb{1}) \text{ or } k = 0 \\ \max(u \times T_c(j), d_k(i,j) - d_{k-1}(i,j)), & \text{if } e_k(i,j) = \mathbb{1} \text{ and } e_{k-1}(i,j) = \mathbb{0} \\ \tau_k(\mathcal{G}_m) + d_{k-1}(i,j)), & \text{if } e_k(i,j) = \mathbb{0} \text{ and } e_{k-1}(i,j) = \mathbb{0} \\ \tau_k(\mathcal{G}_m), & \text{otherwise} \end{cases} \tag{4}$$

where $e(i,j) = \mathbb{0}$ indicates weakly-connected edge, $e(i,j) = \mathbb{1}$ indicates strongly-connected edge; $\tau_k(\mathcal{G}_m)$ is the cycle time at the $k$-th computation round during the training process.

**Cycle Time**. The cycle time per round is the time required to complete a communication round (Marfoq et al., 2020). In this work, we define the cycle time per round is the maximum delay between all silo pairs with strongly-connected edges. Therefore, the average cycle time of the entire training is:

$$\tau(\mathcal{G}_m) = \frac{1}{k} \sum_{k=0}^{k-1} \left( \max_{j \in \mathcal{N}_i^{++} \cup \{i\}, \forall i \in \mathcal{V}} (d_k(j, i)) \right) \tag{5}$$

where $\mathcal{N}_i^{++}$ is an in-neighbors silo set of $i$ whose edges are strongly-connected.

## 4 METHODOLOGY

Our method first constructs the multigraph based on an overlay. Then we parse this multigraph into multiple states that may have isolated nodes. Note that, our method does not choose isolated nodes randomly, but relies on the delay time. In our design, each isolated node has a long delay time in a current communication round. However, in the next round, its delay time will be updated using Eq. 4, and therefore it can become a normal node. This strategy allows us to reduce the waiting time with isolated nodes, while ensuring that isolated nodes can become normal nodes and contribute to the training in the next communication round.

### 4.1 MULTIGRAPH CONSTRUCTION

---

**Algorithm 1:** Multigraph Construction

**Input:** Overlay $\mathcal{G}_o = (\mathcal{V}, \mathcal{E}_o)$;
          Maximum edge between two nodes $t$.
**Output:** Multigraph $\mathcal{G}_m = (\mathcal{V}, \mathcal{E}_m)$;
          List number of edges between silo pairs $\mathcal{L}$.
```
// Compute delay in overlay
```
1   $D_o \leftarrow$ NULL
2   **foreach** $e(i, j) \in \mathcal{E}_o$ **do**
3      $d(i, j) \leftarrow$ Using Eq. 3
4      Append $d(i, j)$ into $D_o$
```
   // Construct multigraph
```
5   $d_{min} = \min(D_0)$ `// find smallest delay`
6   $\mathcal{E}_m \leftarrow$ NULL `// multiset of edges`
7   $\mathcal{L}[|\mathcal{V}|, |\mathcal{V}|] \leftarrow \{0\}$
8   **foreach** $e(i, j) \in \mathcal{E}_o$ **do**
9      $n(i, j) = \min\left(t, \text{round}\left(\frac{d(i,j)}{d_{\min}}\right)\right)$ `// find number of edges for (i,j)`
10     $\mathcal{E}_t \leftarrow$ NULL `// temporary edge set`
11     Append $e(i, j) = \mathbb{1}$ into $\mathcal{E}_t$
12     **foreach** $(n(i, j) - 1)$ **do**
13        Append $e(i, j) = \mathbb{0}$ into $\mathcal{E}_t$
14     Append $\mathcal{E}_t$ into $\mathcal{E}_m$
15     $\mathcal{L}[i, j] \leftarrow n(i, j)$.
16   **return** $\mathcal{G}_m = (\mathcal{V}, \mathcal{E}_m)$; $\mathcal{L}$

---

Algorithm 1 describes our methods to generate the multigraph $\mathcal{G}_m$ with multiple edges between silos. The algorithm takes the overlay $\mathcal{G}_o$ as input. Similar to Marfoq et al. (2020), we use the Christofides algorithm (Monnot et al., 2003) to obtain the overlay.

In Algorithm 1, we focus on establishing multiple edges that indicate different statuses (strongly-connected or weakly-connected). To identify the total edges between a silo pair, we divide the delay $d(i, j)$ by the smallest delay $d_{\min}$ over all silo pairs, and compare it with the maximum number of edges parameter $t$ ($t = 5$ in our experiments). We assume that the silo pairs with longer delay will have more weakly-connected edges, hence potentially becoming the isolated nodes. Overall, we aim to increase the number of weakly-connected edges, which generate more isolated nodes to speed up

the training process. Note that, from Algorithm 1, each silo pair in the multigraph should have one strongly-connected edge and multiple weakly-connected edges. The role of the strongly-connected edge is to make sure that two silos have a good connection in at least one communication round.

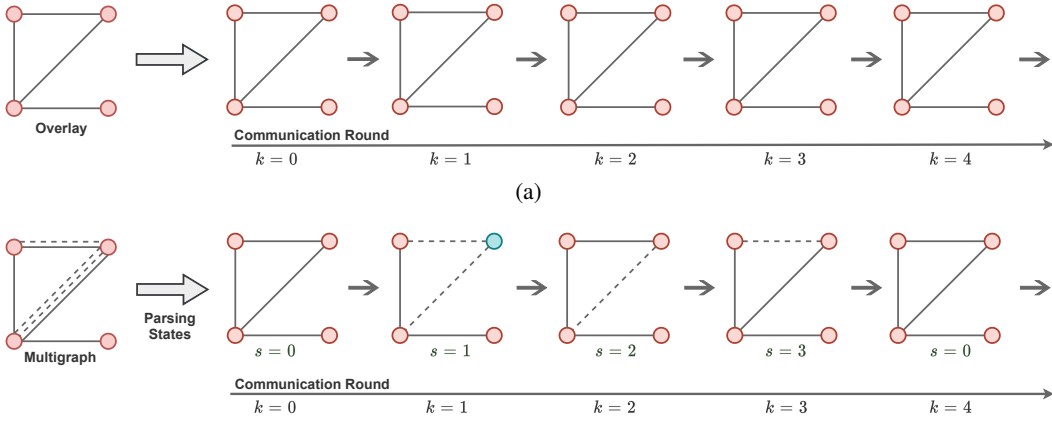

(a)

(b)

Figure 3: The comparison between RING (Marfoq et al., 2020) topology and our multigraph topology in each communication round. (a) RING uses the same overlay in each round. (b) Our proposed multigraph is parsed into different graph states. Each graph state is used in a communication round. Lines denote strongly-connected edges, dotted lines denote weakly-connected ones, and the blue color indicates isolated nodes.

## 4.2 MULTIGRAPH PARSING

In Algorithm 2, we parse multigraph $\mathcal{G}_m$ into multiple graph states $\mathcal{G}_m^s$. Graph states are essential to identify the connection status of silos in a specific communication round to perform model aggregation. In each graph state, our goal is to identify the isolated nodes. During the training, isolated nodes update their weights internally and ignore all weakly-connected edges that connect to them.

To parse the multigraph into graph states, we first identify the maximum of states in a multigraph $s_{\max}$ by using the least common multiple (LCM) (Hardy et al., 1979). We then parse the multigraph into $s_{\max}$ states. The first state is always the overlay since we want to make sure all silos have a reliable topology at the beginning to ease the training. The reminding states are parsed so there is only one connection between two nodes. Using our algorithm, some states will contain isolated nodes. During the training process, only one graph state is used in a communication round. Figure 3 illustrates the training process in each communication round using multiple graph states.

## 4.3 MULTIGRAPH TRAINING

The original DPASGD algorithm (Wang & Joshi, 2018) can not be directly used with our multigraph because the learning process will be terminated when it first meets an isolated node. To overcome this problem, we introduce an upgraded version of DPASGD, namely, DPASGD++ (See Algorithm 3 for details). In each communication round, a state graph $\mathcal{G}_m^s$ is selected in a sequence that identifies the topology design used for training. We then collect all strongly-connected edges in the graph state $\mathcal{G}_m^s$ in such a way that nodes with strongly-connected edges need to wait for neighbors, while the isolated ones can update their models. Formally, the weight in DPASGD++ is updated as:

$$\mathbf{w}_i\left(k+1\right) = \begin{cases} \sum_{j \in \mathcal{N}_i^{++} \cup \{i\}} \mathbf{A}_{i,j} \mathbf{w}_j\left(k-h\right), & \text{if } \mathrm{k} \equiv 0\left(\bmod u+1\right) \& \left|\mathcal{N}_i^{++}\right| > 1, \\ \mathbf{w}_i\left(k\right) - \alpha_k \frac{1}{b} \sum_{h=1}^{b} \nabla L_i\left(\mathbf{w}_i\left(k\right), \xi_i^{(h)}\left(k\right)\right), & \text{otherwise.} \end{cases} \quad (6)$$

where $(k-h)$ is the index of the considered weights; $h$ is initialized to 0 and is changed when the condition in Eq. 7 is met, i.e.,

$$h = h + 1, \quad \text{if } e_{k-h}(i,j) = \mathbb{0} \quad (7)$$

Through Eq. 6 and Eq. 7, at each state, if a silo is not an isolated node, it must wait for the model from its neighbor to update its weight. If a silo is an isolated node, it can use the model in its neighbor from the $(k - h)$ round to update its weight immediately.

---

**Algorithm 2: Multigraph Parsing**

**Input:** Multigraph $\mathcal{G}_m = (\mathcal{V}, \mathcal{E}_m)$;
      List edge numbers between silo pairs $\mathcal{L}$.
**Output:** List of multigraph states
      $\mathcal{S} = \{\mathcal{G}_m^s = (\mathcal{V}, \mathcal{E}_m^s)\}$.

1   $s_{max} \leftarrow \text{LCM}(\mathcal{G}_m)$ (Hardy et al., 1979)
2   $\bar{\mathcal{L}} = \mathcal{L}; \bar{\mathcal{E}}_m^s \leftarrow \text{NULL}$
   // Establish states
3   **for** $s = 0$ **to** $s_{max}$ **do**
4     $\mathcal{E}_t \leftarrow \text{NULL}$ // temporary edge set
5     **foreach** $e(i, j) \in \mathcal{E}_m$ **do**
6       **if** $\bar{\mathcal{L}}[i, j] = \mathcal{L}[i, j]$ **then**
7         Append $e(i, j) = \mathbb{1}$ into $\mathcal{E}_t$
8       **else**
9         Append $e(i, j) = \mathbb{0}$ into $\mathcal{E}_t$
10      **if** $\bar{\mathcal{L}}[i, j] = 1$ **then**
11        $\bar{\mathcal{L}}[i, j] = \mathcal{L}[i, j]$
12      **else**
13        $\bar{\mathcal{L}}[i, j]- = 1$
14     Append $\mathcal{E}_t$ into $\bar{\mathcal{E}}_m^s$
15 **return** $\mathcal{S} = \{\mathcal{G}_m^s = (\mathcal{V}, \mathcal{E}_m^s)\}$ by using $\bar{\mathcal{E}}_m^s$.

---

**Algorithm 3: DPASGD++ Algorithm**

**Input:** List of multigraph states $\mathcal{S}$;
      Initial weight $\mathbf{w}_i(0)$ for each
   silo $i$;
      Maximum training round $K$.

1   $c = 0$ // states counting variable
2   **for** $k = 0$ **to** $K - 1$ **do**
3     $\mathcal{G}_{m_c}^s \leftarrow$ Select $c$-th $\mathcal{G}_m^s$ in $\mathcal{S}$
4     $c = c + 1$
5     **if** $c \geq sizeof(\mathcal{S})$ **then**
6       $c = 0$
7     **for** $i = 0$ **to** $N$ **do**
8       $\mathcal{N}_i^{++} \leftarrow$ strongly-connected edges list of $i$ using $\mathcal{G}_{m_i}^s$.
   // The loop below is parallel
9     **foreach** silo $i \in N$ **do**
10      **for** $\flat = 0$ **to** $u$ **do**
11        $m_\flat \leftarrow$ Sampling from local dataset of $i$
12        $\mathbf{w}_i(k + 1) \leftarrow$ Update model using Eq. 6.

---

## 5 EXPERIMENTS

### 5.1 EXPERIMENTAL SETUP

**Datasets.** We use three datasets in our experiments to evaluate our multigraph topology: Sentiment140 (Go et al., 2009), iNaturalist (Van Horn et al., 2018), and FEMNIST (Caldas et al., 2018). All datasets and the pre-processing process are conducted by following recent works (Wang et al., 2019) and (Marfoq et al., 2020). Details of our experimental setup is in Appendix D.

**Network**. Following Marfoq et al. (2020), we consider five distributed networks in our experiments: Exodus, Ebone, Géant, Amazon (Miller et al., 2010) and Gaia (Hsieh et al., 2017). The Exodus, Ebone, and Géant are from the Internet Topology Zoo (Knight et al., 2011). The Amazon and Gaia network are synthetic and are constructed using the geographical locations of the data centers.

**Baselines**. We compare our multigraph topology with recent state-of-the-art topology designs: STAR (Brandes, 2008), MATCHA (Wang et al., 2019), MATCHA(+) (Marfoq et al., 2020), MST (Prim, 1957), δ-MBST (Marfoq et al., 2020), and RING (Marfoq et al., 2020).

### 5.2 RESULTS

Table 1 shows the cycle time of our method in comparison with other recent approaches. This table illustrates that our proposed method significantly reduces the cycle time in all setups with different networks and datasets. In particular, compared to the state-of-the-art RING (Marfoq et al., 2020), our method reduces the cycle time by 2.18, 1.5, 1.74 times in average in the FEMNIST, iNaturalist, Sentiment140 dataset, respectively. Our method also clearly outperforms MACHA, MACHA(+), and MST by a large margin. The results confirm that our multigraph with isolated nodes helps to reduce the cycle and training time in federated learning.

Table 1: Cycle time (ms) comparison between different typologies. ($\downarrow \circ$) indicates our reduced times compared with other methods.

| Dataset | Network | Topology Design | | | | | | Ours |
|---|---|---|---|---|---|---|---|---|
| | | STAR | MATCHA | MATCHA(+) | MST | $\delta$-MBST | RING | |
| FEMNIST | Gaia | 289.8 ($\downarrow$ 18.5) | 166.4 ($\downarrow$ 10.6) | 166.4 ($\downarrow$ 10.6) | 77.2 ($\downarrow$ 4.9) | 77.2 ($\downarrow$ 4.9) | 57.2 ($\downarrow$ 3.6) | **15.7** |
| | Amazon | 98.8 ($\downarrow$ 7.3) | 57.7 ($\downarrow$ 4.2) | 57.7 ($\downarrow$ 4.2) | 28.7 ($\downarrow$ 2.1) | 28.7 ($\downarrow$ 2.1) | 20.3 ($\downarrow$ 1.5) | **13.6** |
| | Géant | 132.2 ($\downarrow$ 11.0) | 46.9 ($\downarrow$ 3.9) | 102.3 ($\downarrow$ 8.5) | 40.1 ($\downarrow$ 3.3) | 40.1 ($\downarrow$ 3.3) | 27.7 ($\downarrow$ 2.3) | **12.0** |
| | Exodus | 265.2 ($\downarrow$ 21.9) | 84.7 ($\downarrow$ 7.0) | 211.5 ($\downarrow$ 17.5) | 84.4 ($\downarrow$ 7.0) | 84.4 ($\downarrow$ 7.0) | 24.7 ($\downarrow$ 2.0) | **12.1** |
| | Ebone | 190.9 ($\downarrow$ 15.0) | 61.5 ($\downarrow$ 4.8) | 112.6 ($\downarrow$ 8.9) | 60.9 ($\downarrow$ 4.8) | 60.9 ($\downarrow$ 4.8) | 18.5 ($\downarrow$ 1.5) | **12.7** |
| iNaturalist | Gaia | 390.9 ($\downarrow$ 5.7) | 227.4 ($\downarrow$ 3.3) | 227.4 ($\downarrow$ 3.3) | 138.1 ($\downarrow$ 2.0) | 138.1 ($\downarrow$ 2.0) | 118.1 ($\downarrow$ 1.7) | **68.6** |
| | Amazon | 288.1 ($\downarrow$ 3.5) | 123.9 ($\downarrow$ 1.5) | 123.9 ($\downarrow$ 1.5) | 89.7 ($\downarrow$ 1.1) | 89.7 ($\downarrow$ 1.1) | **81.3** ($\downarrow$ 1.0) | **81.3** |
| | Géant | 622.3 ($\downarrow$ 9.1) | 107.9 ($\downarrow$ 1.6) | 452.5 ($\downarrow$ 6.6) | 101 ($\downarrow$ 1.5) | 101 ($\downarrow$ 1.5) | 109 ($\downarrow$ 1.6) | **68.1** |
| | Exodus | 911.9 ($\downarrow$ 14.6) | 145.7 ($\downarrow$ 2.3) | 593.2 ($\downarrow$ 9.5) | 145.3 ($\downarrow$ 2.3) | 145.3 ($\downarrow$ 2.3) | 103.9 ($\downarrow$ 1.7) | **62.6** |
| | Ebone | 901.7 ($\downarrow$ 13.9) | 122.5 ($\downarrow$ 1.9) | 579.9 ($\downarrow$ 8.9) | 121.8 ($\downarrow$ 1.9) | 121.8 ($\downarrow$ 1.9) | 95.3 ($\downarrow$ 1.5) | **64.9** |
| Sentiment140 | Gaia | 323.8 ($\downarrow$ 10.5) | 186 ($\downarrow$ 6.0) | 186 ($\downarrow$ 6.0) | 96.8 ($\downarrow$ 3.1) | 96.8 ($\downarrow$ 3.1) | 76.8 ($\downarrow$ 2.5) | **31.0** |
| | Amazon | 164.6 ($\downarrow$ 4.6) | 79.2 ($\downarrow$ 2.2) | 79.2 ($\downarrow$ 2.2) | 48.4 ($\downarrow$ 1.4) | 48.4 ($\downarrow$ 1.4) | 40.0 ($\downarrow$ 1.1) | **35.8** |
| | Géant | 310.5 ($\downarrow$ 10.3) | 66.6 ($\downarrow$ 2.2) | 222.6 ($\downarrow$ 7.4) | 59.7 ($\downarrow$ 2.0) | 59.7 ($\downarrow$ 2.0) | 54.9 ($\downarrow$ 1.8) | **30.3** |
| | Exodus | 495.4 ($\downarrow$ 17.7) | 104.3 ($\downarrow$ 3.7) | 346.3 ($\downarrow$ 12.4) | 104.1 ($\downarrow$ 3.7) | 104.1 ($\downarrow$ 3.7) | 50.6 ($\downarrow$ 1.8) | **28.0** |
| | Ebone | 444.2 ($\downarrow$ 15.3) | 81.1 ($\downarrow$ 2.8) | 262.2 ($\downarrow$ 9.0) | 80.5 ($\downarrow$ 2.8) | 80.5 ($\downarrow$ 2.8) | 43.9 ($\downarrow$ 1.5) | **29.1** |

From Table 1, our multigraph achieves the minimum improvement under the Amazon network in all three datasets. This can be explained that, under the Amazon network, our proposed topology does not generate many isolated nodes. Hence, the improvement is limited. Intuitively, when there are no isolated nodes, our multigraph will become the overlay, and the cycle time of our multigraph will be equal to the cycle time of the overlay in RING.

## 5.3 ABLATION STUDY

**Convergence Analysis**. Figure 4 shows the training loss versus the number of communication rounds and the wall-clock time under Exodus network using the FEMNIST dataset. This figure illustrates that our proposed topology converges faster than other methods while maintaining the model accuracy. We observe the same results in other datasets and network setups. We provide the proof of convergence of our proposed method in Appendix A.

**Cycle Time and Accuracy Trade-off.** In our method, the maximum number of edges between two nodes $t$ in Algorithm 1 mainly affects the number of isolated nodes. This leads to a trade-off between the model accuracy and cycle time. Table 2 illustrates the effectiveness of this parameter. When $t = 1$, we technically consider there are no weak connections and isolated nodes. Therefore, our method uses the original overlay from RING. When $t$ is set higher, we can increase the number of isolated nodes, hence decreasing the cycle time. In practice, too many isolated nodes will limit the model weights to be exchanged between silos. Therefore, models at isolated nodes are biased to their local data and consequently affect the final accuracy.

**Multigraph vs. RING vs. Random Strategy.** The isolated nodes plays an important role in our method as we can skip the model aggregation step in the isolated nodes. In practice, we can have a trivial solution to create isolated nodes by randomly removing some nodes from the overlay of RING. Table 3 shows the experiment results in two scenarios on FEMNIST dataset and Exodus Network: *i)* Randomly remove some silos in the overlay of RING, and *ii)* Remove most inefficient silos (i.e., silos with the longest delay) in the overlay of RING. Note that, in RING, one overlay is used in all communication rounds. From Table 3, the cycle time reduces significantly when two aforementioned scenarios are applied. However, the accuracy of the model also drops greatly. This experiment shows that although randomly removing some nodes from the overlay of RING is a trivial solution, it can not maintain model accuracy. On the other hand, our multigraph not only reduces the cycle time of the model, but also preserves the accuracy. This is because our multigraph can skip the aggregation step of the isolated nodes in a communication round. However, in the next round, the delay time of these isolated nodes will be updated, and they can become the normal nodes and contribute to the final model.

Table 2: Cycle time and accuracy trade-off with different value of $t$, i.e., the maximum number of edges between two nodes.

| Topology | $t$ | Cycle time (ms) | Acc(%) |
|---|---|---|---|
| RING | _ | 24.7 | 71.05 |
| Multigraph (ours) | 1 | 24.7 | 71.05 |
| | 3 | 13.5 | 71.08 |
| | 5 | 12.1 | 71.13 |
| | 8 | 11.9 | 69.27 |
| | 10 | 11.9 | 69.27 |
| | 20 | 11.9 | 69.27 |
| | 30 | 11.9 | 69.27 |

Table 3: The cycle time and accuracy of our multigraph vs. RING with different criteria.

| Methods | Criteria | #Removed Nodes | Cycle Time (ms) | Acc (%) |
|---|---|---|---|---|
| | Baseline | _ | 24.7 | 71.05 |
| RING | Randomly remove silos in overlay | 1 | 23.1 | 70.63 |
| | | 5 | 21.7 | 68.57 |
| | | 10 | 18.8 | 64.23 |
| | | 20 | 13.0 | 61.2 |
| | Remove most inefficient silos | 1 | 22.5 | 70.71 |
| | | 5 | 19.5 | 68.37 |
| | | 10 | 15.8 | 63.13 |
| | | 20 | 11.2 | 61.48 |
| Multigraph (ours) | _ | _ | **12.1** | **71.13** |

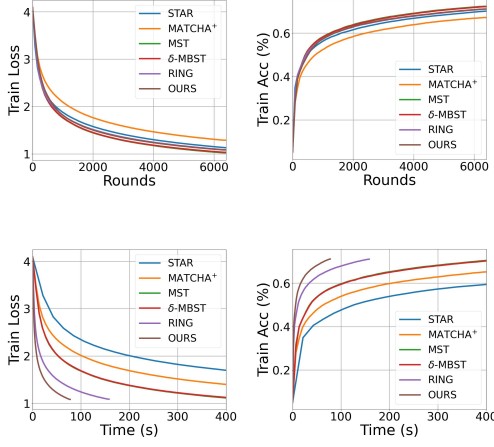

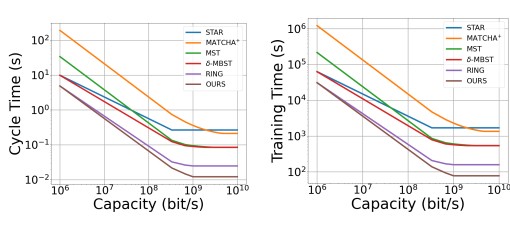

(a) Homogeneous access link capacity

(a) Train Loss  (b) Train Accuracy

(b) 10 Gbps Orchestra access capacity

Figure 4: Convergence analysis of our multigraph under communication rounds (top row) and wall-clock time (bottom row). All access links have the same 10 Gbps capacity. The training time is counted until the training process of all setups finishes $6,400$ communication rounds.

Figure 5: The effect of access link capacity on cycle time and training time of different approaches. (a) All access links have the same 1 Gbps capacity. (b) One orchestra node has a fixed 10 Gbps access link capacity. All setups are trained for $6,400$ communication rounds.

**Access Link Capacities Analysis**. Following Marfoq et al. (2020), we analyse the effect of access link capacity on our multigraph topology. Access link capacity is related to the bandwidth when packages are transmitted between silos. Figure 5 shows the results under Exodus network and FEMNIST dataset in two scenarios: all access links have the same 1 Gbps capacity and one orchestra node has a fixed 10 Gbps access link capacity. From Figure 5, we can see that our multigraph topology slightly outperforms RING when the link capacity is low. However, when the capacity between silos is high, then our method clearly improves over RING. In all setups, our method archives the best cycle time and training time.

## 6 CONCLUSION

We proposed a new multigraph topology for cross-silo federated learning. Our method first constructs the multigraph using the overlay. Different graph states are then parsed from the multigraph and used in each communication round. Our method reduces the cycle time by allowing the isolated nodes in the multigraph to do model aggregation without waiting for other nodes. The intensive experiments on three datasets show that our proposed topology achieves new state-of-the-art results in all network and dataset setups.

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

SUMMARY OF THE APPENDIX

This appendix contains additional materials for the paper "*Multigraph Topology Design for Cross-Silo Federated Learning*". We include mathematical proofs, experimental setup details, additional results, and further discussions. The appendix is organized as follows:

- Section A provides the theoretical proofs to show that our DPASGD++ algorithm converges in all cases.

- Section B illustrates the wall-clock time comparison between our DPASGD++ algorithm and other optimizers.

- Section C discusses the complexity of our proposed Algorithm 1, Algorithm 2, and Algorithm 3 in terms of both theory and practice.

- Section D provides detailed experiment setups in our experiments.

- Section E includes additional experimental results, including accuracy analysis (Section E.1), and additional results on other datasets (Section E.2)

- Section F discusses relevant issues to our work, such as privacy concerns, limitations, and broader impact.

## A  CONVERGENCE ANALYSIS

The decentralized periodic averaging stochastic gradient descent (DPASGD) (Wang & Joshi, 2018) is a popular algorithm for training federated learning setup since it allows local-update in each silo during the learning process. For the convenience, we re-write the equation for updating the weight of silo $i$ in each training round of DPASGD is updated as follows:

$$
\mathbf{w}_i\left(k+1\right) = \begin{cases} \sum_{j \in \mathcal{N}_i^+ \cup \{i\}} \mathbf{A}_{i,j} \mathbf{w}_j\left(k\right), & \text{if k} \equiv 0 \left(\bmod u+1\right), \\ \mathbf{w}_i\left(k\right) - \alpha_k \frac{1}{b} \sum_{h=1}^{b} \nabla L_i\left(\mathbf{w}_i\left(k\right), \xi_i^{(h)}\left(k\right)\right), & \text{otherwise.} \end{cases} \tag{8}
$$

where $b$ is the batch size; $i, j$ denote the siloes, $u$ is the number of local updates, $\alpha_k > 0$ is a potentially varying learning rate at $k$-th round, $\mathbf{A} \in \mathbb{R}^{N \times N}$ is a consensus matrix with non-negative weights, and $\mathcal{N}_i^+$ is the in-neighbors set that silo $i$ has the connection to.

The original DPASGD algorithm can not be directly used with our multigraph because it was not designed to handle the isolated node. We have proposed the DPASGD++ algorithm (Algorithm 3) to overcome this problem.

We next theoretically show in the following propositions that our proposed DPASGD++ is a general case for DPASGD, and will become the original DPASGD with some certain conditions. Therefore, our DPASGD++ will ensure the convergence of the federated training process when using our multigraph topology.

**Proposition 1.** *Assuming that all states of multigraph contains only strongly-connected edge, i.e., $e(i,j) = \mathbb{1}, \forall (i,j) \in \mathcal{G}_m$. Then, DPASGD++ described in Eq. 6 becomes the original DPASGD (Wang & Joshi, 2018).*

*Proof.* In deed, if $e(i,j) = \mathbb{1}, \forall (i,j) \in \mathcal{G}_m$, from Eq. 7 and Eq. 6 we have $h = 0$, and then DPASGD++ becomes the original DPASGD (Wang & Joshi, 2018). □

**Proposition 2.** *Assuming that all states of multigraph contains only weakly-connected edge (all nodes are isolated), i.e., $e(i,j) = \mathbb{0}, \forall (i,j) \in \mathcal{G}_m$ and $\left|\mathcal{N}_i^{++}\right| = 1, \forall i \in \mathcal{G}_m$. Then, DPASGD++ becomes DPASGD (Wang & Joshi, 2018) when DPASGD has $u \to \infty$, and we have*

$$
\mathbf{w}_i\left(k+1\right) = \mathbf{w}_i\left(k\right) - \alpha_k \frac{1}{b} \sum_{h=1}^{b} \nabla L_i\left(\mathbf{w}_i\left(k\right), \xi_i^{(h)}\left(k\right)\right) \tag{9}
$$

*Proof.* Given the assumption that $\left|\mathcal{N}_i^{++}\right| = 1, \forall i \in \mathcal{G}_m$, by combining this with Eq. 6, and noting that the first case in Eq. 6 is violated, we obtain

$$\mathbf{w}_i\left(k+1\right) = \mathbf{w}_i\left(k\right) - \alpha_k \frac{1}{b} \sum_{h=1}^{b} \nabla L_i\left(\mathbf{w}_i\left(k\right), \xi_i^{(h)}\left(k\right)\right).$$

$\square$

## B  WALL-CLOCK TIME ILLUSTRATION

Figure 6 illustrates the wall-lock time that needs to finish the corresponding communication round between our DPASGD++ and other methods. We can see that by utilizing the isolated node, our DPASGD++ takes less time for computation compared to other methods.

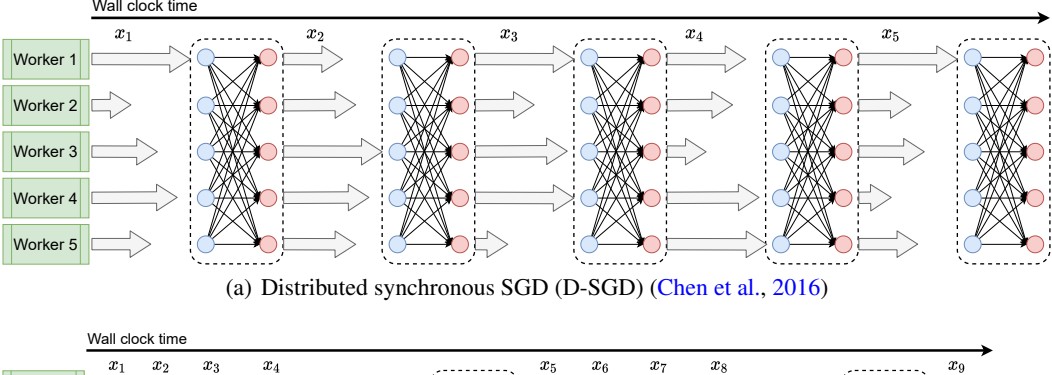

(a) Distributed synchronous SGD (D-SGD) (Chen et al., 2016)

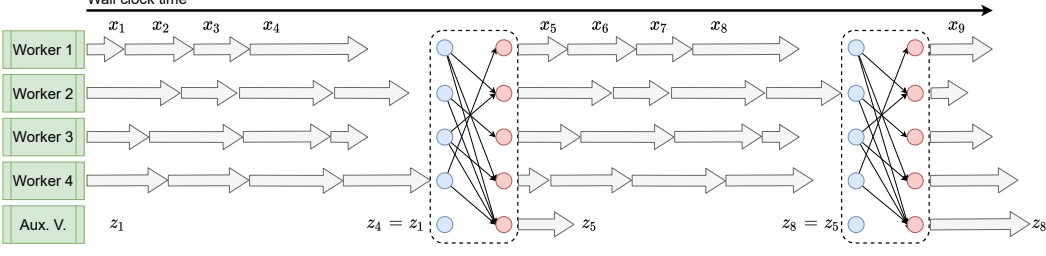

(b) Cooperative SGD (DPASGD) (Wang & Joshi, 2018)

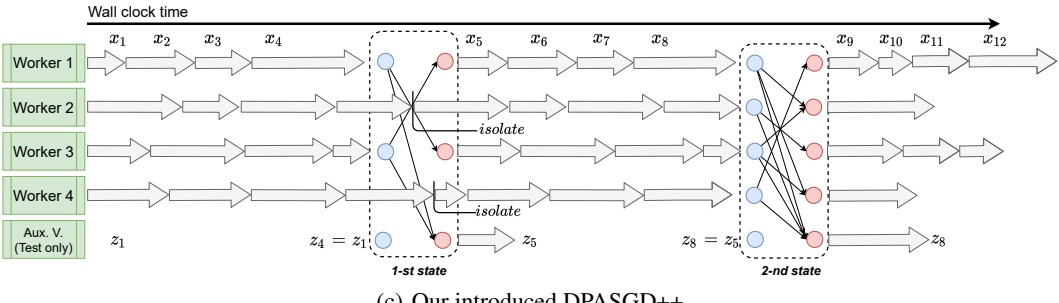

(c) Our introduced DPASGD++

Figure 6: Comparison in wall-clock time between three different learning schemes, including Distributed synchronous SGD (Chen et al., 2016), Cooperative DPASGD (Wang & Joshi, 2018), and our introduced DPASGD++.

## C  ALGORITHM COMPLEXITY

It is trivial to see that the complexity of our proposed Algorithm 1, Algorithm 2, and Algorithm 3 is $\mathcal{O}(n^2)$. In practice, since the cross-silo federated learning setting has only a few hundred silos ($n < 500$) (Kairouz et al., 2019), the time to execute our algorithms is just a tiny fraction of training

time. Therefore, our proposed topology still can significantly reduce the overall wall-clock training time.

## D EXPERIMENT SETUP

**Network**. Table 4 shows the statistic of five distributed networks in our experiments: Exodus, Ebone, Géant, Amazon (Miller et al., 2010) and Gaia (Hsieh et al., 2017). The Exodus, Ebone, and Géant are from the Internet Topology Zoo (Knight et al., 2011). The Amazon and Gaia network are synthetic networks that are constructed using the geographical locations of the data centers.

| Network | #Silos | #Maximum Links |
|---|---|---|
| Gaia (Hsieh et al., 2017) | 11 | 55 |
| Amazon (Miller et al., 2010) | 22 | 231 |
| Géant (Knight et al., 2011) | 40 | 61 |
| Exodus (Knight et al., 2011) | 79 | 147 |
| Ebone (Knight et al., 2011) | 87 | 161 |

Table 4: The network setups in our experiments.

**Datasets.** Table 5 shows the dataset setups in details. We use three standard federated datasets in our experiments, including: Sentiment140 (Caldas et al., 2018), iNaturalist (Van Horn et al., 2018), and FEMNIST (Caldas et al., 2018). All datasets and the pre-processing process are conducted by following Marfoq et al. (2020).

| Dataset | FEMNIST | Sentiment140 | iNaturalist |
|---|---|---|---|
| Task | Image Classification | Sentiment Analysis | Large-scale Image Classification |
| #Samples | 805M | 1,600M | 450M |
| Model | Marfoq' CNN | LSTM | ResNet18 |
| #Params | 1,2M | 4,8M | 11,2M |
| Batch size | 128 | 512 | 16 |
| Model size | 4.62 | 18.38 | 42.88 |

Table 5: Dataset statistic and model details in our experiments. The model size is in Mbits.

**Implementation**. We use PyTorch with the MPI backend in our implementation. The maximum number of edges between two nodes $t$ is set to $5$ in all experiments. Our source code and trained models will be released.

**Hardware Setup**. Since measuring the cycle time is crucial to compare the effectiveness of all topologies in practice, we test and report the cycle time of all baselines and our method on the same NVIDIA Tesla P100 16Gb GPUs. No overclocking is used.

## E ADDITIONAL EXPERIMENT RESULTS

### E.1 ACCURACY ANALYSIS

In federated learning, improving the model accuracy is not the main focus of topology designing methods. However, preserving the accuracy is also important to ensure model convergence. Table 6 shows the accuracy of different topologies after $6,400$ communication training rounds on the FEMNIST dataset. This table illustrates that our proposed method achieves competitive results with other topology designs.

Table 6: Accuracy comparison between different topologies. The experiment is conducted using the FEMNIST dataset. The accuracy is reported after $6,400$ communication rounds in all methods.

| Network | Topology | | | | | |
|---------|------|-----------|-----|--------|------|------|
|         | **STAR** | **MATCHA(+)** | **MST** | **δ-MBST** | **RING** | **Ours** |
| Gaia | 69.09 | 68.43 | 68.86 | 68.95 | 68.2 | 68.45 |
| Amazon | 69.59 | 69.06 | 69.65 | 70.37 | 69.78 | 69.63 |
| Géant | 68.91 | 65.57 | 69.44 | 68.94 | 69.3 | 68.98 |
| Ebone | 69.66 | 64.48 | 71.91 | 70.62 | 70.29 | 70.23 |
| Exodus | 70.14 | 67.21 | 72.36 | 72.19 | 71.05 | 71.13 |

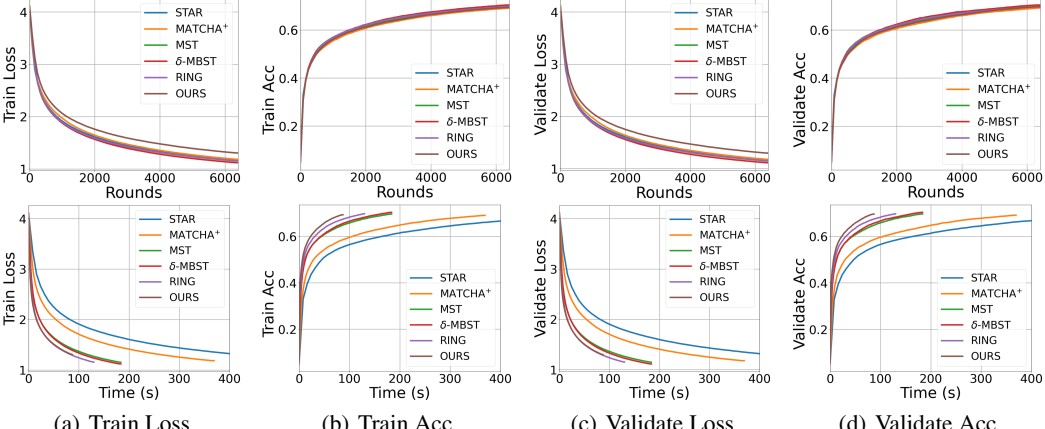

(a) Train Loss     (b) Train Acc     (c) Validate Loss     (d) Validate Acc

Figure 7: The comparison of our multigraph and other methods on the convergence w.r.t. communication rounds (top row) and wall-clock time (bottom row). Amazon network and FEMNIST dataset are used. The wall-clock time is counted until the training process of all setups reaches $6,000$ communication rounds. Best viewed in color.

### E.2 ADDITIONAL RESULTS

Figure 7, Figure 8, and Figure 9 show more experiment results of our multigraph and other methods on different datasets and network setups. These figures show that our multigraph achieves state-of-the-art accuracy compared to other methods (top row) while significantly reducing the total wall-clock time (bottom row).

## F FURTHER DISCUSSION

**Limitations**. Since our multigraph is designed based on RING (Marfoq et al., 2020) overlay, our method inherits both the strengths and weaknesses of RING. We can see that the "lower bound" of our multigraph is the overlay of RING when there are no isolated nodes. In this case, all states in our multigraph are the input overlay. Hence, there is no improvement. Furthermore, compared to RING, our multigraph is more sensitive to the low bandwidth capacity setup (Figure 5).

**Privacy Concerns.** The privacy of federated learning has been discussed in many papers (Li et al., 2021a; Kairouz et al., 2019). Generally, any federated learning topologies (such as ours) can be accompanied by any privacy preservation methods (Marfoq et al., 2020). Since our method focuses on optimizing topology, studying the privacy concern is not our main focus. In practice, our proposed topology potentially can enhance privacy as it reduces the data exchanges between nodes.

**Broader Impacts**. Decentralized inference mechanisms are a workable alternative that balances utility with crucial normative values like privacy, transparency, and accountability. Given the ease of access to sensitive data, large-scale centralized data sources are not feasible to uphold the best interests and privacy of the users. We hope that using our proposed topology, we can reduce the

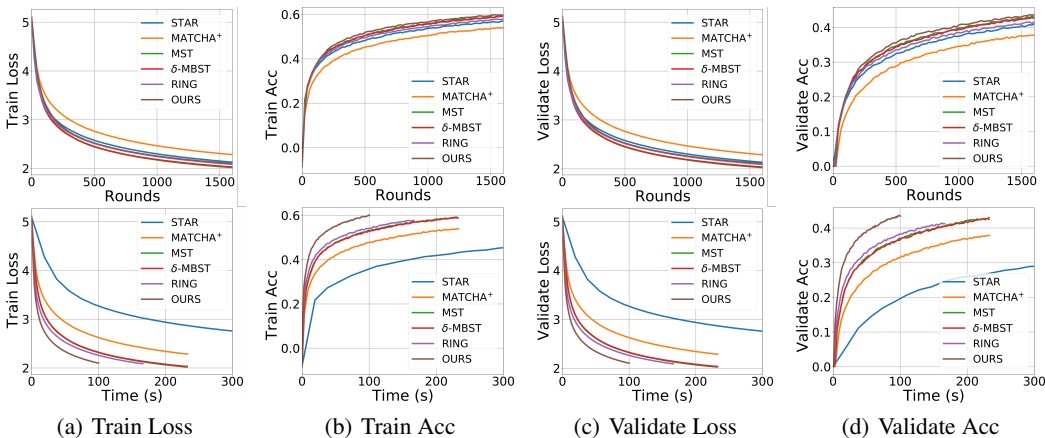

Figure 8: The comparison of our multigraph and other methods on the convergence w.r.t. communication rounds (top row) and wall-clock time (bottom row). Exodus network and iNaturalist dataset are used. The wall-clock time is counted until the training process of all setups reaches $1,500$ communication rounds. Best viewed in color.

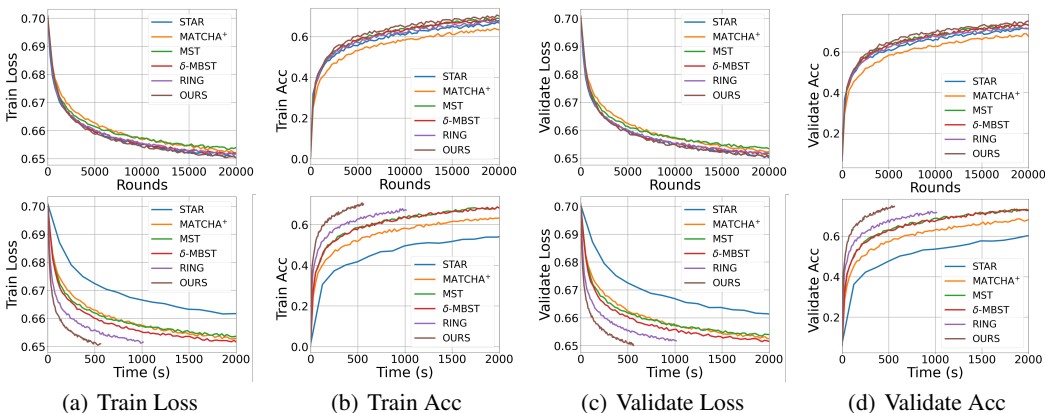

Figure 9: The comparison of our multigraph and other methods on the convergence w.r.t. communication rounds (top row) and wall-clock time (bottom row). Exodus network and Sentiment140 dataset are used. The wall-clock time is counted until the training process of all setups reaches $20,000$ communication rounds. Best viewed in color.

training time of the whole network, and consequently would bring direct benefits in real-world applications such as saving energy when training a model, reducing waiting time, and potentially reducing the privacy breach when exchanging data between silos.

