# OpenReview forum: "Multigraph Topology Design for Cross-Silo Federated Learning"
_ICLR.cc/2023/Conference — Submitted to ICLR 2023_

### Official Review · Reviewer_QFKN · 2022-10-24

**Confidence:** 3
**Clarity, Quality, Novelty And Reproducibility:** See the above comments
**Correctness:** 3
**Technical Novelty And Significance:** 2
**Empirical Novelty And Significance:** 2
**Recommendation:** 3

**Strength And Weaknesses:**

- Strength: This is an interesting problem, especially the dynamic topology part which is under-explored.
- Weakness:
   - The paper is poorly written and to some extent not self contained. Many of the pieces necessary for developing the method/algorithm are defined or presented without context. This makes the paper difficult to follow. As an example I was not able to follow why eq. 4 holds and the explanations around it was definitely not sufficient.
  - In algorithm 1, it is not clear what is the exact motivation behind the algorithm, what are the constraints the extracted topologies need to satisfy, and how do the authors know that it does exactly what is designed to do. It is presented just as is without much context. The same is also true (to a lesser extent) about Algorithm 2.
  - It is not clear if such dynamic topological changes would still allow the overall optimization algorithm to converge to a unified solution over the network? In fact, the interplay between the optimization method and dynamical topology which seems to be an important aspect of such algorithms is not even discussed in the paper. How do the algorithms scale with the number of participating silos? do global models converge? what are the conditions needed for the model to converge correctly? And are they satisfied by the proposed algorithms?
  - In the experiments, the results only show the train loss and train accuracy, while for ML models the test loss and test accuracy are more important. Also, regarding the distributed training procedure it is not clear what train loss and train accuracies are reported? Are these measured at the nodes (with local models) and then averaged? or are they with the global model? Why there are no error bars reported? Are the results averaged over multiple runs?

**Summary Of The Paper:**

This paper considers designing dynamic network topologies to improve the training time and performance of the cross silo FL training. The authors propose an algorithm and verify its effectiveness through empirical evaluation.

**Summary Of The Review:**

Overall, the presentation of the paper makes it difficult to comprehensively judge the quality of the paper and the proposed method. In order to improve the paper, I strongly suggest the authors improve the presentation, make the paper more self-contained and give more details about the algorithm designs, limitations and the interplay between the optimization and dynamic topology. Moreover, a more rigorous experimental section is also needed.

---

> ### Author Response · Authors · 2022-11-18
> **Replies for comments of Reviewer QFKN.**
>
> We thank the reviewer for your insightful feedback. We would like to explain our paper in more detail:
>
> **Writing of our paper:** We are sorry to hear that you think our paper is difficult to follow. In fact, our paper mainly focuses on the construction and parsing of the multigraph to generate isolated nodes. During each training round, we can skip the model aggregation step on the isolated nodes, hence reducing the training time for FL.
>
> **Equation 4 concern:** Equation $4$ is our proposed method to calculate the delay time. We update the delay time in the current training round based on the delay time of the previous training round, depending on whether the edge is weakly or strongly-connected.
>
> **Algorithm 1 concern:** In Algorithm $1$, we want to construct the multigraph based on the overlay. We rely on the delay time and the hyper-parameter $t$ (i.e., the maximum number of edges between two nodes) to construct the multigraph. The key idea is that the node with a long delay will cause problems since other nodes need to wait for it. Therefore, we assume that the silo pairs with longer delay will have more weakly-connected edges, hence potentially becoming the isolated nodes.
>
> **Algorithm 2 concern:** In Algorithm $2$, we parse the multigraph into graph states. Please see Figure $3$ (bottom row) for an illustration.
>
> **Converge concern:** We extend the DPASGD algorithm in Algorithm $3$ for training our method. The convergence proof is provided in Section A of our Supplementary Material. We note that our paper focuses on cross-silo training, which only has a few hundred reliable data silos. DPASGD still guarantees convergence with many more silos, however, this is not the setup we have in practice during training with cross-silo FL.
>
> **Experiments setup and accuracy:** We strictly follow the most recent baseline RING paper (Marfoq et al., 2020) in our experiment. The results of other state-of-the-art methods were also taken from the RING paper. Due to the page limitation, we present the accuracy in our Supplementary Material (Table $6$, Figure $7$, Figure $8$, Figure $9$). We discuss the trade-off between training time and accuracy in Table $2$ of our main paper.

---

### Official Review · Reviewer_rd7C · 2022-10-27

**Confidence:** 3
**Correctness:** 3
**Technical Novelty And Significance:** 3
**Empirical Novelty And Significance:** 3
**Recommendation:** 6

**Clarity, Quality, Novelty And Reproducibility:**

Clarity: The paper is well organized.

Quality: The proposed method is well analyzed and discussed with detailed experiments.

Novelty: The paper uses multigraph to model a very scarce scenario on federated learning.


**Strength And Weaknesses:**

Strength.
1. Using multigraph topology in federated learning is a novel idea.
2. This paper is well written.

Weaknesses:
1. The targeting scenario is very narrow. It requires a cross-silo federated setting with a peer-to-peer network and a graph topology.
2.  It is unclear In the main result of table 1, accuracy results should also be provided to show that the proposed algorithm could achieve comparable accuracy with less training time.
3. Below paper could be discussed in the related work. It is a similar idea to graph-based aggregation on federated learning [1].

[1] Personalized Federated Learning with A Graph


**Summary Of The Paper:**

This paper proposed a new multigraph topology for cross-silo federated learning in a decentralized network. Specifically, the proposed method extends the overlay graph with weak connections based on the delay time and improved DPASGD algorithm to process isolated nodes for time reduction.

**Summary Of The Review:**

The paper is organized very well. The paper’s targeting problem is very scarce, thus the contribution to the federated learning community is limited. The novelty is sourced from the combination of multigraph and federated learning.

---

> ### Author Response · Authors · 2022-11-18
> **Replies for comments of Reviewer rd7C.**
>
> We thank the reviewer for the constructive comments. We would like to clarify as follows:
>
> **Targeting scenario:** We agree that the scenario requires a peer-to-peer network and a graph topology. However, this is not the problem of our paper as this is the setup of the field, and all papers on this direct follow this setup.
>
> **Accuracy results:** Our paper mainly focuses on training time, so we present Table 1 with training time only (and also due to page limitation). We also present the accuracy in our Supplementary Material (Table $6$, Figure $7$, Figure $8$, Figure $9$). We discuss the trade-off between training time and accuracy in Table 2 of our main paper.
>
> **Related work:** We thank the reviewer for the reference. We will discuss it in our next version.

---

### Official Review · Reviewer_7Rwz · 2022-10-30

**Confidence:** 5
**Correctness:** 3
**Technical Novelty And Significance:** 3
**Empirical Novelty And Significance:** 3
**Recommendation:** 5

**Clarity, Quality, Novelty And Reproducibility:**

This paper introduces some novel techniques but many places are lack of clarity as I mentioned in the last section.

**Strength And Weaknesses:**

**Strength**
- The proposed topology design based on multigraphs is novel. It may have great impact on future research on this topic.
- The experiments are kind of comprehensive, which include three different datasets and five different communication networks.


**Weakness**
- Clarity on key insights
    - While I feel the proposed algorithm in this paper looks interesting and novel, and the experimental results are promising, I found that it is hard to get the key ideas behind the algorithm. If I understand correctly, the benefits of the proposed algorithm lie in two aspects: (1) at each round, the activated graph is just a subgraph of the overlay, thus there are less communication links; (2) for the isolated nodes, we will not let them continue performing local training. Instead, they will average with the latest available neighbor models. This can help to improve the convergence. Unfortunately, these two benefits are not clearly stated in the paper. And I believe there might be more benefits hidden in the paper. Even for an expert, it may be difficult to effectively get what are the key insights behind the algorithm.
- Clarity on technical details: I found that the authors may omit a lot of technical details or explanations.
    - It is unclear to readers how you derived equation 4. It is hard to interpret it without any explanations. For example, what does it mean by "$d_k(i,j) = d_k(i,j)$" on the first line. Do you update $d_k(i,j)$ based on its previous values at $k-1$, $k-2$?
    - The authors mentioned that in order to overcome the problem of DPASGD, they propose some new techniques. However, how and why these techniques can overcome the problem are unclear.
    - Related to the above point, it would be better to clearly state why DPASGD has the problem. The authors mentioned that DPASGD will terminate when a node is isolated. As far as I know, this is not true, as long as the expected graph over all rounds is connected. Then, DPASGD won't have the divergence problem.
    - How do you measure the cycle time? If you want to measure the real time, then you must have the same number of machines as the number of silos. On some networks, there is about 80 silos. Does that mean you ran experiments on a cluster of 80 machines? If not, the reported numbers will be just simulation results. The authors should provide full details on how they simulate the numbers.
    - It is unclear how you implemented the MATCHA algorithm. Basically, MATCHA has a hyper-parameter to tune the communication time per round. How did you choose this hyper-parameter?
- Incomplete study: I felt there are still many questions remaining. The authors are supposed to address them in order to make the paper complete and self-contained.
   - The authors just stated the steps of the multi-graph parsing algorithm. But we should also know its parsing properties. For example, as the authors mentioned, "Using our algorithm, some states will contain isolated nodes." It would be better to show either theoretically or empirically how the exact number of states have isolated nodes in different type of graphs. Does this number depend on the number of edges, maximal degree, or something else? Answering these questions would help us to better understand the benefits of the proposed algorithm and when is proper to apply it.
   - Similarly, we readers know little things about the DPASGD++ algorithm. The authors claim that it ensures model convergence. However, this claim is not supported in the main paper. Ideally, one should have a convergence analysis to support this claim. The authors provided one in the Appendix, which is very weak. It says DPASGD++ is equivalent to DPASGD when all states are strong edges and all states are weak edges. This conclusion is obvious and is not interesting. If it is equivalent to DPASGD, then there is no need to have a new algorithm. Therefore, the interesting part is something in between. What happens when some states are strong and others are weak. Without a convergence analysis, it is possible that DPASGD++ can just diverge in some cases.


**Summary Of The Paper:**

This paper studies how to speed up cross-silo federated learning, in which different silos need to have multiple peer-to-peer communication/synchronization at each round. Following previous works, this paper proposes a new efficient communication topology based on multigraph. Extensive experiments on multiple datasets validate the effectiveness of the proposed method.

**Summary Of The Review:**

I like the proposed algorithm and it seems that it can be an important baseline in the future. But I do think the current draft is not ready to be published. Both the key insights and technical details are incomplete and unclear. The authors are supposed to perform more studies around the proposed algorithm. More details can be found in the weakness section.

---

> ### Author Response · Authors · 2022-11-18
> **Replies for comments of Reviewer 7Rwz.**
>
> We thank the reviewer for the immensely insightful comments. We would like to clarify as follows:
>
> **Clarity on key insights**
> - The key idea of our paper is the use of multigraph to create isolated nodes. During each training round, we can skip the model aggregation step on the isolated nodes, hence reducing the training time. We mentioned this key idea several times in our paper.
> - The isolated nodes are created dynamically and depend on the number of weakly-connected edges. We rely on the fact that the node with a long delay will cause problems since other nodes need to wait for it. Therefore, we assume that the silo pairs with longer delay will have more weakly-connected edges, hence potentially becoming the isolated nodes. We mentioned this idea in the explanation of Algorithm $1$ at the end of page $5$.
>
> **Clarity on technical details**
> - **Equation 4: Delay time:** Yes, we update the delay time based on the delay time of the previous round. $d_k(i,j) = d_k(i,j),  if  \left(e_k(i,j) = 1 \text{ and } e_{k-1}(i,j) = 1 \right)\text{ or } k=0$ simply means the delay time of current round is similar to previous round if $e_k(i,j)$ is the strongly connected edge, and $e_{k-1}(i,j)$ is also a strongly-connected edge, or k=0 which is the  first communication round.
> - **DPASGD in the Multigraph:** We agree that DPASGD won't have the divergence problem as long as the expected graph over all rounds is connected and in our multigraph case, it is simply inapplicable to use directly rather than causing divergence. Obviously, the connected graph is also the premise of the DPASGD which can not be directly achieved in multigraph since a graph state contains isolated nodes and may generate more than one connected graph in a single communication round. Naively using DPASGD may let strong nodes deadlock to listening to the isolated nodes or listen to nodes from other connected graphs. DPASGD++ is just a modification of DPASGD so as to make the learning process able to achieve within each connected graph (or each isolated node) in each state.
> - **Circle time:** The circle time is defined in Equation $5$ of our paper.
> - **Experiment setup, simulation, and MATCHA**: We strictly follow the RING paper (Marfoq et al., 2020) in our setup for a fair comparison,  including the simulation provided in their codes. We also do not tune the MATCHA algorithm. The results of MATCHA in our paper were taken from the RING paper.
>
> **Incomplete study:**
> - **Isolated nodes:** The numbers of isolated nodes vary based on the network configuration (Amazon, Gaia, Exodus,...), the parameter $t$ (maximum number of edges between two nodes), and the delay time which is identified by many factors (geometry distance, model size, computational cost based on tasks, bandwidth,...). For more details about the factors of delay time, please visit Equations $3$ and $4$. We agree that factors that form isolated nodes need more intensive investigation. However, this is the first paper that introduces the application of a dynamic graph in decentralized federated learning. Comprehensive studies of these factors for isolated nodes should be put in future work so as to make sure the analysis is good enough in terms of qualitative and quantitative. At this time, we provide empirical experiments about the rounds and states that have the appearance of isolated nodes. Hope that it can help readers understand more about the effectiveness of isolated nodes.
>
> | Network | Total silos | $t$ | rounds with isolated nodes/ max training rounds | states with isolated nodes/ max possible states | Cycle time (s) and amount of speed up (times) |
> |:---:|:---:|:---:|:---:|:---:|:---:|
> | Gaia | 11 | 5 | 4693/6400 | 44/60 | 15.7 ($\downarrow$3.6) |
> | Amazon | 22 | 5 | 2133/6400 | 2/6 | 13.6 ($\downarrow$1.5) |
> | Géant | 40 | 5 | 4266/6400 | 8/12 | 12.0 ($\downarrow$2.3) |
> | Exodus | 79 | 5 | 3306/6400 | 31/60 | 12.1 ($\downarrow$2.0) |
> | Ebone | 87 | 5 | 2346/6400 | 11/30 | 12.7 ($\downarrow$1.5) |
>
> We conduct experiments on the FEMNIST dataset in terms of five network configurations (Gaia, Amazon, Geant, Exodus, Ebone). All experiments are trained with $6400$ communication rounds and are controlled with a maximum of $5$ edges between two nodes in multigraph ($t =  5$). We then record the states and rounds that have the appearance of isolated nodes. As we can see, the speed-up effect increase when more rounds/ states that have isolated nodes have been found.
> - **Convergence of DPASGD++**: It is easy to see that, there are $2$ scenarios that can happen if there are some isolated nodes in a specific graph state: (i)  the isolated nodes visible in the state have their convergence ability similar to the lower bound case, and (ii) connected graphs visible in the state have the convergence ability follow the original DPASGD.

---

### Decision · Program_Chairs · 2023-01-20

**Decision:**

Reject

**Justification For Why Not Higher Score:**

- Missing in-depth study about key properties of the proposed method, like number of isolated nodes and its effect on convergence
- Many unanswered questions of reviewers, like "How do the algorithms scale with the number of participating silos? do global models converge? what are the conditions needed for the model to converge correctly? And are they satisfied by the proposed algorithms?"

**Justification For Why Not Lower Score:**

N/A

**Metareview: Summary, Strengths And Weaknesses:**

The paper attempts to improve distributed learning in cross-silo setting. In this regard, the authors propose a new dynamic communication topology for cross-silo federated learning which is claimed to reduce the training time and improve performance. The key reason for the efficiency is the existence of "isolated nodes" in the dynamic topology which allows model aggregation to happen without waiting for other nodes. Experiments are conducted on five simulated distributed networks and three datasets to show effectiveness of the proposed method. All reviewers agree that studying dynamic topology is interesting, but feel like current work is somewhat incomplete. We thank the authors for detailed response, but unfortunately it still left many questions remained. The paper can be made more self-contained by providing further analysis (either theoretical or empirical) of the two algorithms presented in the paper. In particular, emergence of isolated nodes is crucial for success of the proposed method, but it is not analyzed like what are the guarantees or expected number of states that have isolated nodes in different type of graphs. Moreover, effect of these isolated nodes on convergence is not studied, only a very weak result is provided in the appendix. Adding the study of interplay between the optimization method and dynamical topology which seems to be an important aspect of proposed method would make the paper very interesting and useful to the community. Making the baselines, like MATCHA, stronger by adapting them to studied setting will more strongly showcase the improvement of the proposed and further strengthen the paper. Finally, for some reviewers/readers improving writing by providing a detailed motivation might be helpful.

**Summary Of Ac-Reviewer Meeting:**

N/A